# Manganese Porphyrin Promotes Post Cardiac Arrest Recovery in Mice and Rats

**DOI:** 10.3390/biology11070957

**Published:** 2022-06-24

**Authors:** Peng Wang, Ying Li, Baihui Yan, Zhong Yang, Litao Li, Zhipeng Cao, Xuan Li, Ines Batinic-Haberle, Ivan Spasojevic, David S. Warner, Huaxin Sheng

**Affiliations:** 1Multidisciplinary Neuroprotection Laboratories, Center of Perioperative Organ Protection, Department of Anesthesiology, Duke University Medical Center, Durham, NC 27710, USA; kylinwp@163.com (P.W.); petal216@163.com (Y.L.); ybh258404025@126.com (B.Y.); yangzhong305@126.com (Z.Y.); dingding51800@163.com (L.L.); zpcao@cmu.edu.cn (Z.C.); lixuanzhengding@163.com (X.L.); david.warner@duke.edu (D.S.W.); 2Department of Anesthesiology, The Fifth Central Hospital of Tianjin, Tianjin 300450, China; 3Department of Cardiology, The Fifth Central Hospital of Tianjin, Tianjin 300450, China; 4Department of Anesthesiology, The Second Affiliated Hospital of Xi’an Jiaotong University, Xi’an 710000, China; 5Department of Orthopedics, The Fifth Central Hospital of Tianjin, Tianjin 300450, China; 6Department of Neurology, Hebei General Hospital, Shijiazhuang 050051, China; 7School of Forensic Medicine, China Medical University, Shenyang 110122, China; 8Department of Anesthesiology, Harbin Medical University Cancer Hospital, Harbin 150081, China; 9Department of Radiation Oncology, Duke University Medical Center, Durham, NC 27710, USA; ibatinic@duke.edu; 10Department of Medicine, Duke University Medical Center, Durham, NC 27710, USA; ivan.spasojevic@duke.edu; 11Pharmacokinetics/Pharmacodynamics Core Laboratory, Duke University School of Medicine, Durham, NC 27710, USA; 12Department of Surgery, Duke University Medical Center, Durham, NC 27710, USA; 13Department of Neurobiology, Duke University Medical Center, Durham, NC 27710, USA

**Keywords:** cardiac arrest, functional deficit, manganese porphyrin, BMX-001, ischemia/reperfusion, outcome

## Abstract

**Simple Summary:**

Although heart failure is a long-term chronic disease, it can also lead to arrhythmia, triggering cardiac arrest, and eventually death. Our study is a randomized and blinded preclinical assessment of the efficacy of the Mn porphyrin compound, MnTnBuOE-2-PyP^5+^ (BMX-001), commonly known as a superoxide dismutase mimic, in both mouse and rat models of cardiac arrest and resuscitation. Our study indicates that MnTnBuOE-2-PyP^5+^ improves survival, neurologic function recovery, and behavioral performance in animals suffering from cardiac arrest. The results support the development of Mn Porphyrin-based therapeutics for the treatment of heart failure and cardiac arrest.

**Abstract:**

*Introduction* Cardiac arrest (CA) and resuscitation induces global cerebral ischemia and reperfusion, causing neurologic deficits or death. Manganese porphyrins, superoxide dismutase mimics, are reportedly able to effectively reduce ischemic injury in brain, kidney, and other tissues. This study evaluates the efficacy of a third generation lipophilic Mn porphyrin, MnTnBuOE-2-PyP^5+^, Mn(III) *ortho meso-*tetrakis (*N*-n-butoxyethylpyridinium-2-yl)porphyrin (MnBuOE, BMX-001), in both mouse and rat models of CA. *Methods* Forty-eight animals were subjected to 8 min of CA and resuscitated subsequently by chest compression and epinephrine infusion. Vehicle or MnBuOE was given immediately after resuscitation followed by daily subcutaneous injections. Body weight, spontaneous activity, neurologic deficits, rotarod performance, and neuronal death were assessed. Kidney tubular injury was assessed in CA mice. Data were collected by the investigators who were blinded to the treatment groups. *Results* Vehicle mice had a mortality of 20%, which was reduced by 50% by MnBuOE. All CA mice had body weight loss, spontaneous activity decline, neurologic deficits, and decreased rotarod performance that were significantly improved at three days post MnBuOE daily treatment. MnBuOE treatment reduced cortical neuronal death and kidney tubular injury in mice (*p* < 0.05) but not hippocampus neuronal death (23% MnBuOE vs. 34% vehicle group, *p* = 0.49). In rats, they had a better body-weight recovery and increased rotarod latency after MnBuOE treatment when compared to vehicle group (*p* < 0.01 vs. vehicle). MnBuOE-treated rats had a low percentage of hippocampus neuronal death (39% MnBuOE vs. 49% vehicle group, *p* = 0.21) and less tubular injury (*p* < 0.05) relative to vehicle group. *Conclusions* We demonstrated the ability of MnBuOE to improve post-CA survival, as well as functional outcomes in both mice and rats, which jointly account for the improvement not only of brain function but also of the overall wellbeing of the animals. While MnBuOE bears therapeutic potential for treating CA patients, the females and the animals with comorbidities must be further evaluated before advancing toward clinical trials.

## 1. Introduction

Cardiac arrest is a serious condition resulting from severe coronary artery disease [1], heart failure [2,3], hemorrhagic shock [4,5], asphyxia [6], overdose of drugs [7,8], and abnormal potassium [9]. During cardiac arrest, organs are subjected to severe ischemic insult due to an abrupt loss of the heart’s ability to pump [10,11,12,13,14,15,16,17]. Although timely cardiopulmonary resuscitation, the use of an automated external defibrillator, and extracorporeal membrane oxygen (ECMO) have significantly improved the survival rate [18,19], the outcome is still poor [20,21]. Survivors suffer from persistent neurologic and psychiatric deficits [21]. Emerging evidence has demonstrated that ischemic brain injury is closely associated with high mortality and poor recovery in cardiac arrest [22,23]. Therefore, we have sought a novel therapeutic approach to ameliorate the ischemic injury and improve the outcomes.

Reactive species (i.e., reactive oxygen, nitrogen, and sulfur species) are formed in the brain during reperfusion post ischemia [24,25,26,27]. The burst in these reactive species is linked to other pathologic changes such as inflammatory responses and apoptotic neuronal death. Oxidative stress and inflammation are considered therapeutic targets for preventing secondary brain injury [28]. Manganese porphyrins are a class of compounds characterized by a redox-active transitional metal (Mn) with a cyclic porphyrin core ligand and side-chain substitutions that tune their redox properties to act as potent superoxide dismutase mimics and peroxynitrite decomposition catalysts. As insights into the redox biology of cells and redox chemistry of Mn porphyrins have greatly increased over the last couple of decades, we have learned that these compounds, which are the most powerful SOD mimics, are also the most able to affect cellular signaling and, in turn, apoptotic and proliferative pathways [29,30]. Briefly, rather than acting as SOD mimics, such Mn porphyrins oxidatively modify activities of transcription factors such as NF-кB and Nrf2 and other critical signaling proteins such as mitogen-activated protein kinases as well as endogenous antioxidative defenses (such as glutathione-*S*-transferase and peroxiredoxins). The modification of proteins occurs via H_2_O_2_-driven catalysis of protein cysteine oxidation [29,30]. Animal studies of brain ischemia have demonstrated that Mn porphyrins attenuate the formation of reactive species [31,32,33], in turn inhibiting the activation of transcription factor NF-кB and expression of inflammatory genes [33,34,35] and upregulate endogenous antioxidative genes via activation of the Nrf2 pathway [36]. Such actions of Mn porphyrins contribute to greatly reduced neurologic deficits and infarct size [30,31,32,33,35]. Another SOD mimic, a para-analog MnTM-4-PyP, also attenuates inflammatory responses in ischemic acute kidney injury [37]. The current study aims to investigate the preclinical efficacy of Mn(III) *ortho meso*-tetrakis(*N*-n-butoxyethylpyridinium-2-yl)porphyrin (MnTnBuOE-2-PyP^5+^, MnBuOE) [38]—a Mn porphyrin combining superior redox-based potency and lipophilicity with low toxicity—in both mice and rat models of cardiac arrest and resuscitation, and to demonstrate its therapeutic potential in suppressing CA injury.

## 2. Materials and Methods

The Duke University Animal Care and Use Committee approved the following experiments. Twenty-four male young C57BL/6J mice (8–10 weeks old) and 20 Wistar rats (250–275 g, 8–9 weeks old) were purchased from Jackson laboratory (Bar Harbor, ME, USA) and Envigo (Indianapolis, IN, USA), respectively, and then housed at the Duke animal facility with free access to food and water and a 12-h light–dark cycle. Room temperature and humidity were well-controlled.

### 2.1. Cardiac Arrest and Resuscitation Model

Animals fasted with free access to water the night before surgery. Prior to surgery, animals were anesthetized with 5% isoflurane in 30% oxygen balanced with nitrogen, intubated orally, and ventilated. Rectal temperature was maintained at 37.0 ± 0.2 °C using a heating lamp and blanket during surgery and ischemia. Three electrical leads were attached to the front paws and left hindleg respectively for electrocardiogram (EKG) monitoring. At surgery, the animals were placed in the supine position and the neck incision was cut to expose the right jugular vein. Then, a PE-10 tube for mice or silicone tube for rats was placed. After 15 min of physiological stabilization, cardiac arrest (CA) was induced by solution of potassium chloride (KCl) as described previously [39,40]. In brief, 300 µL or 5 mL venous blood was withdrawn from the jugular vein in mice or rats into the heparin syringe. Then, 30 µL or 1 mL/kg 0.5 M cold potassium chloride solution was rapidly infused in mice or rats. The EKG flattened immediately following infusion. The timer was then started to observe the CA onset. The ventilator and isoflurane were turned off. Body temperature was allowed to change spontaneously. Blood withdrawal prior to the KCl infusion was performed to reduce the circulating blood volume in the heart; in turn, less volume of KCl was required to successfully flatten the EKG. At three minutes after the CA onset, the withdrawn blood was slowly reinfused to dilute the heart KCl concentration. Then, the syringe for intravenous infusion of epinephrine (32 µg/mL) was connected to the jugular vein catheter. At 8 min post-CA onset, 100% oxygen was made available via ventilator, followed first by a bolus of epinephrine (100 µL in mice and 500 µL in rats) and then by its continuous infusion at 2.5 µL/min. Chest compressions were performed at 300 strokes per minute until the regular sinus rhythm was observed. The maximal volume for epinephrine was 250 µL for mice and 2 mL for rats. The jugular vein tube was then removed, and the skin incision closed. Animals were disconnected from the ventilator after complete restoration of spontaneous respiration and returned to the home cage.

An arterial line was established to measure blood pressure in the rat experiment. A PE-50 tube was inserted into the tail artery. Mean blood pressure was recorded before CA, during CA, and 20 min post CA. The tube was removed after surgery, and the skin incision closed.

### 2.2. Group Assignment and Treatment

Following resuscitation, animals were randomly assigned to vehicle or MnBuOE treatment using the online software GraphPad QuickCalcs (n = 10 in the CA vehicle group and n = 10 in the CA MnBuOE treatment group for each species, 20 mice and 20 rats total). The sample size was calculated using a power analysis based on the neuroscore data from a previous study [39]. In mice, an additional 4 mice received sham surgery and vehicle treatment. The injective MnBuOE solution was prepared using 0.9% sodium chloride, so the vehicle group received 0.9% sodium chloride. The first dose of MnBuOE, 0.1 mg/kg for mice and 0.225 mg/kg for rats, was intraperitoneally given immediately after resuscitation. Subcutaneous injections were then administered twice per day for 3 days in mice (at 0.2 mg/kg/day) and 7 days in rats (at 0.45 mg/kg/day). The vehicle group received the same volume of 0.9% sodium chloride only.

### 2.3. Outcome Measurements

*Body weight* Soft food was prepared in a plastic petri dish daily and placed on the bottom of the cage to allow easy access for the animals. Body weight was recorded daily.

*Spontaneous activity* Mice were gently placed inside a Smart cage (Afasci Research laboratories, Redwood City, CA, USA) and allowed to move freely for 10 min. The time of spontaneous walking in the cage was automatically recorded.

*Neuroscore N*eurologic deficits were examined in mice 3 days post CA and in rats 1 and 7 days after CA using the scoring system as previously reported [39]. The scoring system includes the ability to climb on a vertical screen, walk on a horizontal wooden rod, and hang on horizontal rope. Each performance was scored as 0–3 points based on the time recorded. A total score was based on the data from 3 performances (9 points = normal and 0 = severe injury).

*Rotarod latency* Both mice and rats were examined prior to surgery and 3 days after surgery. Rats survived 7 days and were also examined on day 7. Animals were placed on the accelerated rod (0–40 turns per minute, ENV-577M and ENV-577, Med Associates Inc., Georgia, Vermont), and the latency to fall from the rolling rod was recorded. Three trials with an interval of 15 min were completed on the testing day, and the best performance from 3 trials was used for data analysis. All animals had a training trial, and the performance baseline prior to surgery was determined.

*Histologic Analysis* After behavioral tests, animals were anesthetized with isoflurane and intubated. Intra-cardiac perfusion was performed with heparinized saline followed by 10% formalin. The brain, kidneys, and heart (in rats) were harvested, paraffin-embedded, sliced at a thickness of 5 μm, and stained using Hematoxylin and Eosin (H&E) or Celestine blue and Acid Fuchsin. The cortical damage was graded using a crude damage index with a scale of 0–3 (0, no neuronal damage; 1, less than 30% of neurons were damaged; 2, 30–60% of neurons were damaged; and 3, more than 60% of neurons were damaged) [41]. The number of dead and alive neurons in the entire hippocampal CA1 area was counted, and the percentage of neuronal cell death calculated using the formula CA1 neuronal death (%) = the number of dead neurons/(the number of dead cells + number of alive cells) × 100%. The kidneys were also harvested to assess tubular injury (i.e., loss of brush border, tubular cell lysis, and sloughed debris in the tubular lumen space) from the HE-stained paraffin sections based on the score system (0, no damage seen; 1, 0–25% of tubules were damaged; 2, 25–50% of tubules were damaged; 3, 50–75% of tubules were damaged; 4, more than 75% of tubules were damaged) [42]. Five kidney cortical areas were evaluated to obtain an average score. The left ventricle wall thickness was measured. Three measurements were taken from each slide, and the average measure of wall thickness was used for statistical analysis.

All data were collected by researchers who were blind to the group assignment.

### 2.4. Statistical Analysis

The data are expressed as mean ± SD and were analyzed using unpaired Student’s t-test or two-way ANOVA with repeated measures, except the cortical damage score and kidney tubular injury score, which were nonparametric, and are expressed as median ± IQR, and were analyzed using the Mann–Whitney U test. All statistical analyses were performed using Prism 6 software (GraphPad Software Inc., San Diego, CA, USA), and a *p* value less than 0.05 was considered statistically significant.

## 3. Results

### 3.1. Mouse Study

Most of the mice survived for three days except two vehicle- and one MnBuOE-treated mouse, which died on day 3. The mortality was 20% in the vehicle and 10% in the MnBuOE groups (Figure 1A). All CA mice had a significant body weight loss on days 1, 2, and 3 compared to the sham group (Figure 1B, *p* < 0.01). MnBuOE-treated mice started to gain body weight on day 2, which improved on day 3 (weight loss 18% in MnBuOE vs. 25% in vehicle group, *p* < 0.05). Post CA, these mice were not as active as normal, and spontaneous activity gradually decreased. Both vehicle and MnBuOE groups had a spontaneous activity decline (Figure 1C, *p* < 0.01 vs. sham). The mice treated with MnBuOE had significantly improved activity on day 3 compared to the vehicle mice (Figure 1D, action time 0.382 ± 0.479 min in vehicle, 1.892 ± 1.481 min in MnBuOE group, *p* < 0.05). On day 3, the mice treated with MnBuOE performed better on the screen, bar, and rope, and had a total score of 5.0 ± 5.0 (Figure 1E, *p* < 0.01 vs. 0.5 ± 5.25 in vehicle, median ± IQR). Prior to injury, all mice were trained on the rolling rod, and the latencies to fall from the rod were 262 ± 42, 290 ± 21, and 281 ± 23 s in vehicle, MnBuOE, and sham groups, respectively. CA caused a dramatical drop in the latency to fall that was 3.875 ± 4.883 s in vehicle and 87.778 ± 93.118 s in MnBuOE mice (Figure 1F, *p* < 0.0001 vs. sham 297.25 ± 5.5 s). Rotarod performance was greatly improved in MnBuOE-treated mice (*p* < 0.01).

The global brain ischemia/reperfusion event that occurred during cardiac arrest induced neuronal death, especially in hippocampal CA1 area. The vehicle group had a high percentage of hippocampal CA1 neuronal death that was slightly reduced in MnBuOE treatment (Figure 2A,B, 34.60 ± 32.38% in vehicle, 23.05 ± 34.92% in MnBuOE, *p* = 0.49). The ischemia-induced neuron death was also found in the cortical area and the score obtained was based upon the percentage of dead neurons as 1 (1–30%), 2 (30–60%), and 3 (>60%) [41]. The vehicle group had a score of 1.5 ± 1.75, and the MnBuOE group had a score of 1 ± 1.5 (Figure 2C,D, *p* < 0.05). Ischemic changes were also found in kidneys. The kidney tubular injury score was 1.855 ± 0.08 in vehicle and 1.758 ± 0.18 in MnBuOE (Figure 2E,F, *p* = 0.04). The score was averaged from five different kidney cortical areas. CA-induced histology was reduced in MnBuOE mice.

### 3.2. Rat Study

Mean blood pressure was not different between groups (Figure 3, *p* = 0.71). Rapid infusion of cold KCl solution induced an immediate cardiac arrest, and blood pressure dropped to zero, which was gradually raised after successful resuscitation. Epinephrine infusion caused a large blood pressure increase at 6 min post resuscitation (103 ± 18 mmHg in vehicle, 97 ± 20 mmHg in MnBuOE, *p* = 0.47), which was transiently above the baseline (80 ± 6 mmHg in vehicle, 81 ± 11 in MnBuOE, *p* = 0.65), and decreased at 20 min (54 ± 11 mmHg in. vehicle, 56 ± 8 mmHg in MnBuOE, *p* = 0.68).

Similar to what was found in the mouse study, cardiac arrest and resuscitation also resulted in a sickness in all rats, including loss of body weight and decrease in motor activity. For the first two days, they did not eat and drink enough. Starting from the third day, they gradually gained body weight (Figure 4A). The rats treated with MnBuOE recovered faster than the vehicle group and had an average gain in weight of 20 ± 9 g vs. 3 ± 24 g in vehicle group (*p* < 0.05). Prior to CA, both groups had similar performances on screen, rod, and rope, with no total-score difference (Figure 4B, 9 ± 1 in vehicle and MnBuOE). At 24 h post CA, both groups had a significant performance decline compared to that prior to CA (*p* < 0.01). However, the rats treated with MnBuOE had a better score (5.5 ± 3) than the vehicle group (4.0 ± 3, *p* = 0.01). This difference was absent on day 7 as both groups recovered. All rats had three days of training on the rotarod. The latency data collected at the last day before surgery were used as a baseline for rotarod performance (Figure 4C, the latency to fall 237 ± 87 in vehicle and 233 ± 68 in MnBuOE). On day 3 post CA, both groups had decreased latency compared to baseline (140 ± 71 in vehicle and 171 ± 70 in MnBuOE group). After one week of treatment, the MnBuOE group performed much better on a rotarod when compared to the vehicle group (173 ± 59 in MnBuOE vs. 113 ± 61 in vehicle, *p* = 0.04).

The left ventricular wall thickness was measured to represent contraction ability, a factor that affects blood pressure. The MnBuOE group had a slight increase in wall thickness, which did not reach a significant difference when compared to the vehicle group (Figure 5, 23.6 ± 2.5 mm in MnBuOE vs. 21.0 ± 0.9 mm in vehicle, *p* = 0.35).

The MnBuOE group also had a low percentage of neuronal death in hippocampal CA1 area (Figure 6A,B, 39.66 ± 12.56% in MnBuOE vs. 49.58 ± 20.03% in vehicle, *p* = 0.21). Same pattern was found in CA-induced cortical damage (Figure 6C,D, 1.2 ± 0.1 in MnBuOE vs. 1.6 ± 0.2 in vehicle, *p* = 0.13). The tubular injury was found to be more severe in the vehicle group (Figure 6E,F, 1.5 ± 0.5 in MnBuOE vs. 2.0 ± 0.5 in vehicle, *p* = 0.047), and ischemic renal damage was reduced in the MnBuOE-treated rats.

## 4. Discussion

Cardiac arrest and resuscitation resulted in whole-body ischemia. In addition to the brain, all other organs, including intestine, liver, and kidney, were affected. Thus, when a novel compound effectively protects multiple organs, the effects on general health will be superimposed and robustly improved. Body weight is by far the most direct parameter reflecting the status of the severity of injury and progress in recovery. In both experiments, the animals lost weight on the first two days and started to re-gain the weight on day 3. The mice and rats had a significant body weight increase following MnBuOE treatment. More importantly, the survival rate also improved. Mortality was 20% in the vehicle mice, and that was reduced by 50% when MnBuOE treatment was given. These data objectively demonstrate the protective effect of MnBuOE in this CA model.

Post-CA spontaneous activity, the animal’s ability to perform on screen, rod, rope, and the rolling rod, were also closely associated with MnBuOE treatment. These data were further supported by the data on MnBuOE-driven improved organ functions in CA animals. Brain histology showed that neuronal damage in the cortex and hippocampus was reduced by MnBuOE treatment. The tubular injury in kidney was improved as well. For patient care, survival and functional recovery are the most critical outcomes of cardiac arrest. Thus, our current assessment of MnBuOE potency in a CA model has laid a solid foundation and justifies further studies that will also include female animals and animals with comorbidities.

Multiple studies have proven the protective effect of Mn porphyrin compounds in animal models of ischemic brain injury. These include manganese (III) *meso*-tetrakis(*N*-ethylpyridinium-2-yl) porphyrin (MnTE-2-PyP^5+^, BMX-010) [31], manganese (III) *meso*-tetrakis(*N,N’-*1,3-diethylimidazolium-2-yl) porphyrin (Mn(III)TDE-2-ImP^5+^, AEOL10150) [32,35], and Mn(III) *N*-n-hexylpyridinium-2-yl)porphyrin, (MnTnHex-2-PyP^5+^) [33]. Protection was seen when the treatment began as late as 6 h after the onset of ischemia [31,33]. We have demonstrated that these redox-active drugs are powerful superoxide dismutase mimics, peroxynitrite decomposition catalysts, and regulators of cellular transcriptional activity. Numerous studies in our lab and others have demonstrated their therapeutic potential to treat a variety of human diseases that have oxidative stress in common [30]. Our group and others have reported the impressive efficacy of such redox-active Mn porphyrins in ischemia and reperfusion [43]. Experimental studies of brain ischemia have demonstrated that ischemia and reperfusion induce reactive oxygen and nitrogen species [31], inflammatory genes [34], cytokines [33], and transcription factor NF-kB [33,35], which were all reduced by Mn porphyrin compounds.

In treated CA mice and rats, there was an increase in levels of reactive species due to the injury, including H_2_O_2_. Mn porphyrins use H_2_O_2_ and glutathione to oxidatively modify cysteines of NF-кB and the Keap1 factor of Nrf2. That would in turn inhibit NF-кB and inflammation. The oxidation of Keap1 by Mn porphyrin releases Nrf2 into the nucleus, which would result in upregulation of endogenous antioxidative defenses, such as MnSOD, catalase, peroxiredoxins, and glutathione-S-transferase. These endogenous antioxidants suppress oxidative stress in the organs of CA animals. Huo et al. showed that cardiac arrest induces oxidative stress and inflammation [44]. Serum levels of oxidative stress products (8-iso prostaglandin F2 alpha and MDA) and inflammatory cytokines (TNF alpha, IL-1 beta, and HMGB1) are increased at 3, 12, and 24 h after cardiac arrest [44]. Endogenous SOD and catalase enzymes are decreased [44]. Such efficacy was supported by favorable pharmacokinetic studies in which Mn porphyrins were found to accumulate in different tissue and cellular organelles including mitochondria. The lipophilic MnBuOE and MnTnHex-2-PyP^5+^ were found not only in the heart and brain but also in the mitochondria of these tissues. Both compounds accumulate at equal mitochondria/cytosol ratios of 3.1 in the heart and 2 in the brain [45]. While hydrophilic MnTE-2-PyP^5+^ accumulates at a low 1.6 mitochondria/cytosol ratio in the heart, it was not found in brain mitochondria [45].

Another less potent superoxide dismutase mimetic, due to the pyridyl substituents located in inferior para (4) positions [29,30,45], Mn(III) *meso-*tetrakis(*N*-methylpyridinium-4-yl)porphyrin (MnTM-4-PyP^5+^) has been reported to decrease inflammatory cytokines in acute renal ischemia and reperfusion injury [37]. The serum creatinine was reduced as well. Kidney histology in our experiment, which showed that CA-induced tubular injury was attenuated in MnBuOE-treated animals, is consistent with the report on MnTM-4-PyP^5+^.

The limitation of our study is that we did not examine CA-induced ischemic damage in other organs, such as lungs, intestines, and hearts. However, based on our previous findings [30,45], we believe that MnBuOE should have also improved their functions, thereby contributing to overall post-CA animal recovery. Future studies will clarify the anticipated, full benefit of MnBuOE including in vitro mechanistic investigations.

## 5. Conclusions

This study demonstrated that MnBuOE treatment significantly improved post-CA survival and functional recovery in both rat and mouse models, indicating the therapeutic potential of MnBuOE in protecting organs against ischemia/reperfusion damage. Further studies are needed to assess these effects in females, aged animals, and animals afflicted with comorbidities to provide a complete set of data before advancing MnBuOE to clinical trials.

## Figures and Tables

**Figure 1 biology-11-00957-f001:**
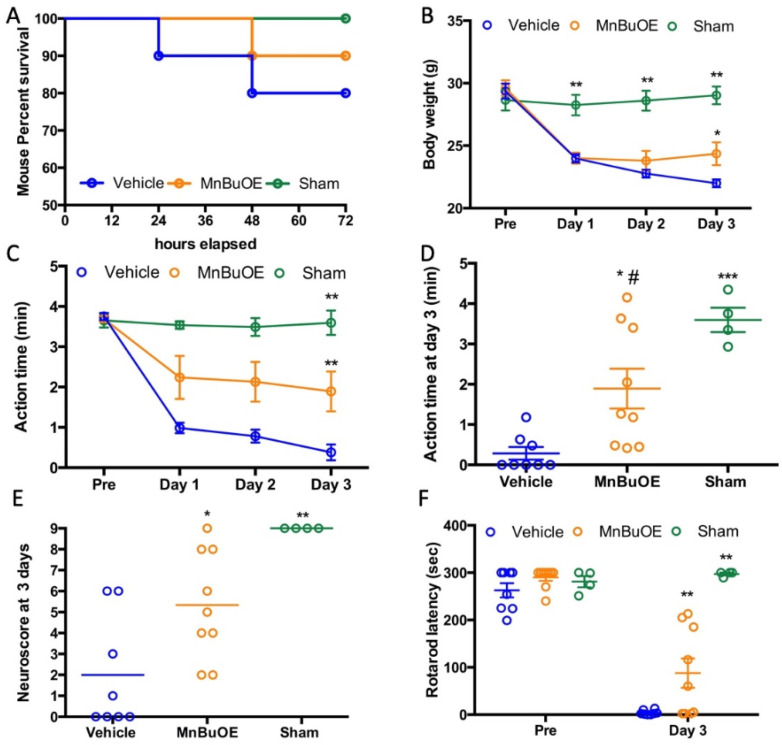
Body weight and functional recovery in mouse that underwent CA. (**A**) Mouse survival curve. (**B**) Daily body weight changes, (**C**) Daily spontaneous activity time, (**D**) Spontaneous activity time on day 3 for individual mice. (**E**) Neuroscore at day 3 post CA. (**F**) Rotarod performance prior to and at day 3 post CA. Data were expressed as mean ± SEM except neuroscore (Bar = median). Each circle represents an individual mouse. Vehicle n = 8, MnBuOE n = 9 and sham n = 4. * *p* < 0.05 vs. vehicle. ** *p* < 0.01 vs. vehicle, *** *p* < 0.001 vs. Vehicle, # *p* < 0.05 vs. Sham.

**Figure 2 biology-11-00957-f002:**
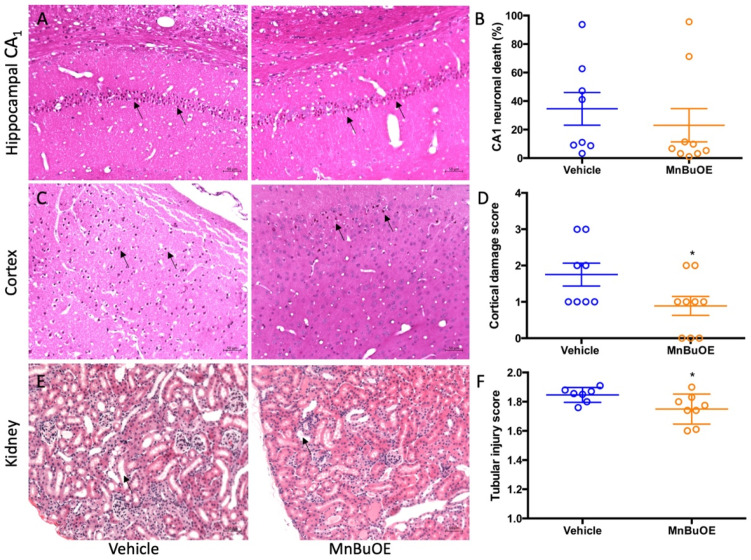
Quantified post-CA histological damage in mice. (**A**) A representative hippocampal CA1 area in CA mice. Arrows point to dead neurons. (**B**) The percentage of CA1 neuronal death in both groups. (**C**) A representative cortex area in CA mice. Arrows point to dead neurons. (**D**) Cortex damage scores in both groups. (**E**) A representative kidney cortex. Arrow points to thinned tubular wall and enlarged tubular cavity. (**F**) Tubular injury score in both groups. Circles represent an individual mice and bars indicate mean ± SEM. * *p* < 0.05 vs. vehicle.

**Figure 3 biology-11-00957-f003:**
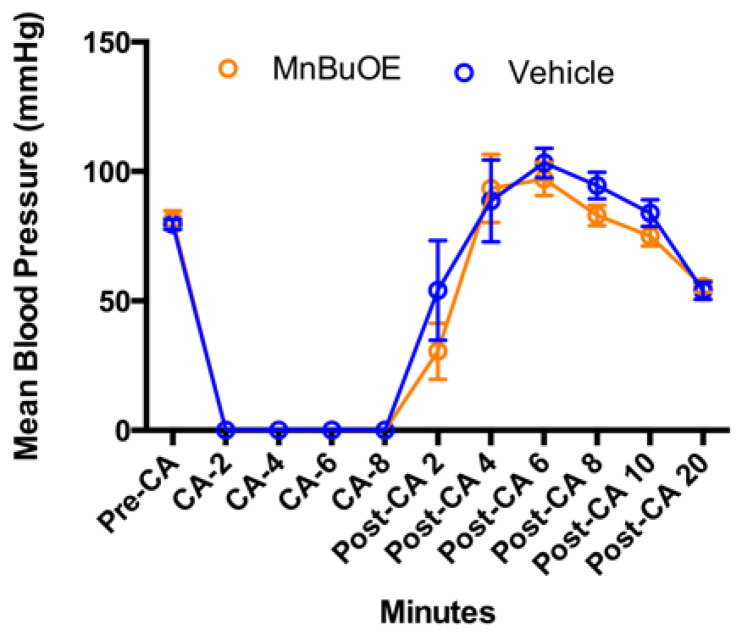
Arterial blood pressure changes in CA rats. Data were collected before CA, during CA, and 20 min after resuscitation in both groups. Data = mean ± SEM.

**Figure 4 biology-11-00957-f004:**
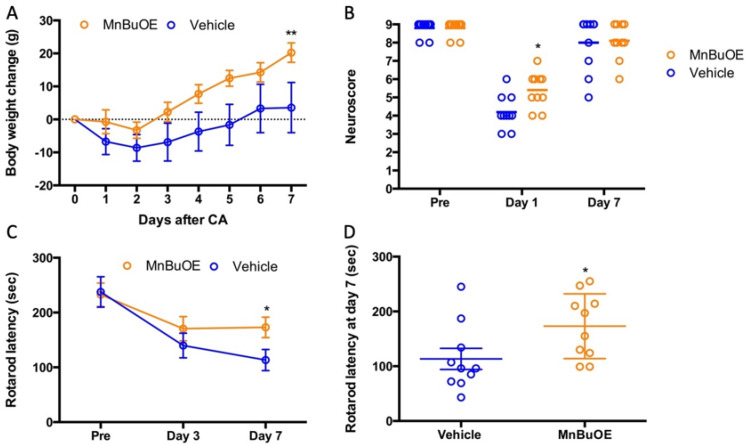
Body weight and neurologic recovery in rat which underwent CA. (**A**) Daily body weight change, (**B**) Neuroscore prior to CA, and at days 1 and 7 post-CA, (**C**) Rotarod performance prior to CA, and at days 1 and 7 post-CA. (**D**) Rotarod performances at day 7 for individual rats. Data were expressed as mean ± SEM except neuroscore. Each circle represents an individual rat. Bar = median. Vehicle n = 10 and MnBuOE n = 10. * *p* < 0.05 vs. vehicle, ** *p* < 0.01 vs. vehicle.

**Figure 5 biology-11-00957-f005:**
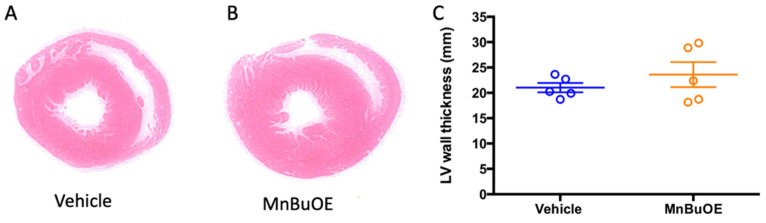
Left ventricular wall thickness at 7 days post-CA. (**A**) a representative left ventricular wall in vehicle rat, (**B**) a representative left ventricular wall in MnBuOE rat, (**C**) left ventricular wall thickness in vehicle (n = 5) and MnBuOE (n = 5). Data were expressed as mean ± SEM (lines in C). Each circle represents an individual rat.

**Figure 6 biology-11-00957-f006:**
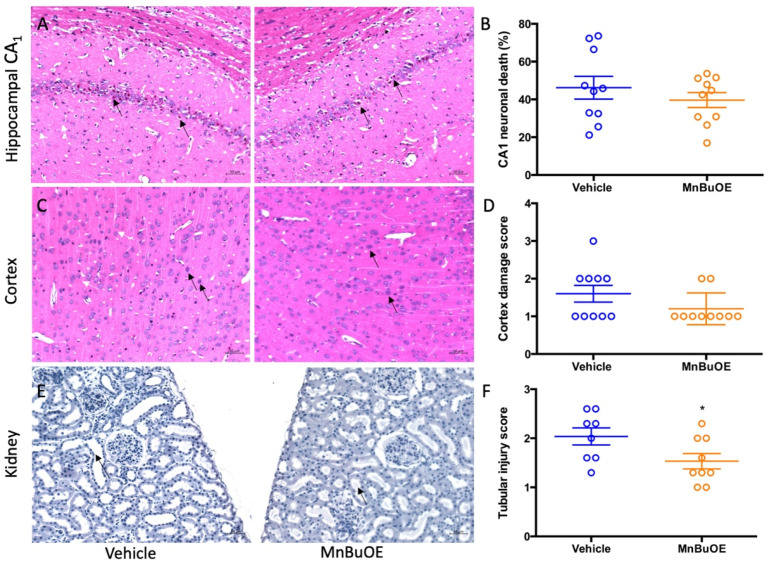
Quantified post-CA histologic damage in rats. (**A**) A representative hippocampal CA1 area in CA rat. Arrows point to dead neurons. (**B**) The percentage of CA1 neuronal death in both groups. (**C**) A representative cortex area in CA rat. Arrows point to dead neurons. (**D**) Cortex damage scores in both groups. (**E**) A representative tubular ischemic injury in CA rat. Arrow points to thinned tubular wall and enlarged tubular cavity. (**F**) Tubular injury score in both groups. Circles represent individual animals and bars indicate mean ± SEM. * *p* < 0.05.

## Data Availability

Data are stored at Duke box and available when requested.

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
