# Peer review of "Manganese Porphyrin Promotes Post Cardiac Arrest Recovery in Mice and Rats"

_biology, 2022, doi:10.3390/biology11070957_

Round 1

Reviewer 1 Report

The study is very interesting.

The methodological design is good, but some improvements are necessary:

1. Numbers of animals used should be reported in detail and adequately explained (i.e refer to a power analysis etc.).

2. Use sham operated animals in both mice and rats.

3. Examine kidney lesions in both mice and rats.

4. Is the estimation of cell death sound? Your microphotographs are not convincing.

Are there no studies from other research groups available on the subject?

I think self-citation of this extend (62,5%) is excessive!

Author Response

Please see attached responses

Reviewer 2 Report

The manuscript entitled „Manganese Porphyrin Promotes Post Cardiac Arrest Recovery in Mice and Rats” by Peng Wang et al, investigates the ability of MnBuOE to improve the post-CA survival, and the functional outcomes in both mice and rats which jointly account for the improvement, not only of brain function but of the overall wellbeing of animals. For this purpose, the authors assessed the animals' BW, spontaneous activity, neuro score, and rotarod latency and also analyzed the brains and kidneys of CA animals using the histological technique.

In general, the experiments were well conducted in this manuscript, still, some aspects need to be clarified.

1.      Line 23. Please provide the full name for Mn porphyrin compound.

2.      Lines 56-59. Please provide some references for this sentence.

3.   Line 66. What do the authors mean when they say reactive species? Is it reactive oxygen species?

4.   Lines 93-99. Please provide the age of Wistar rats and also the number of animals included in the experiment.  

5.  Lines 127-135. Please mention the number of animals per experimental group. Why do the authors use both experimental animal models?

6.      What are the levels of circulating biomarkers in CA mice and rats (oxidative, inflammatory, etc.)?

7.   Did the authors measure the blood pressure of MnBuO-treated animals compared to the vehicle?

8.  The authors mentioned in the introduction section that Mn porphyrins oxidatively modify the activities of NF-KB, Nrf2, and MAP kinases. How do the authors think the expression of these factors is in the brains and kidneys of treated mice and rats versus controls? Also how MnBuOE would influence the expression of these molecules in the heart of CA animals?

9.  Are there any studies investigating the in vitro effect of MnBuEO on cell cultures? experiments performed on cell cultures (renal, neuronal, cardiac cells) would have been a plus for the study.

1Figure 3. Please provide histological images for MnBuEO-treated CA mice compared to vehicles.

1Figure 4. Please provide histological images for MnBuEO-treated CA rats compared to vehicles.

1A simple hematoxylin-staining of the heart sections for MnBuEO-treated CA mice compared to vehicles might represent proof of CA induction and the recovery after CA. the authors should provide these histological images if they exist. 

Author Response

Please see attached responses

Round 2

Reviewer 1 Report

Not including sham operated rats in your study is not right and additionally it makes your study look unbalanced, between mice and rats. Your answer that that was due to short of funding is not very convincing. Now the study seems more complete. I hope that my suggestions helped in that